# Subjective experiences of cognitive decline and receiving a diagnosis of dementia: qualitative interviews with people recently diagnosed in memory clinics in the UK

Penny Xanthopoulou,[ORCID][1] Rose McCabe[2]

[1]Medical School, University of Exeter, Exeter, UK
[2]School of Health Sciences, City, University of London, London, UK

**Correspondence to**
Dr Penny Xanthopoulou;
p.d.xanthopoulou@exeter.ac.uk

## ABSTRACT

**Objectives** To explore people's experiences of cognitive decline and receiving a diagnosis of dementia.

**Design** 61 semistructured interviews within 2 weeks of diagnosis. Audio recordings were transcribed, line-by-line coded using NVIVO V.11 and analysed using thematic analysis.

**Setting** 9 memory clinics (UK).

**Participants** People with mild/moderate dementia.

**Results** Most participants were diagnosed with Alzheimer's disease (56% female, mean age 81 years). 104 codes were grouped into 22 categories, feeding into 9 subthemes and 4 overarching themes: (1) dissonance, threat to identity and visibility of dementia: dementia was associated with a progressive loss of competence, culminating in being an idiot, crazy and losing the plot. The stigma of dementia led people to hide their diagnosis from others, even close family members. However, decreasing competence in everyday tasks was becoming increasingly visible in family and wider social networks. (2) Vulnerability and being in limbo: people were frustrated by the impact of dementia on their lives and felt vulnerable. Moreover, people were disturbed by not knowing how much and when they would deteriorate further. (3) Loss of control and agency: loneliness, increasing dependence and becoming a burden foreshadowed increasing diminished personal agency. (4) Maintaining agency and self-worth: some people focused on what they could do and the benefits of diagnosis. This involved accepting the diagnosis, adapting to changes by using coping strategies and accepting support from others. This helped people to maintain personal agency and self-worth.

**Conclusion** While personal acceptance of dementia is challenging, people are additionally troubled about disclosing their diagnosis to others. Limited time in diagnostic appointments and limited postdiagnostic support leave few opportunities to address the emotional impact of a dementia diagnosis. There may be opportunities for healthcare professionals to discuss with patients the benefits of staying positive, implementing coping strategies and accepting support to live well with dementia.

### Strengths and limitations of this study

► This was a large study involving 61 in-depth interviews with people with dementia within 2 weeks of receiving a diagnosis.

► Participants were from diverse social backgrounds and a range of urban and rural areas in the UK.

► People who did not speak English and those with severe communication problems were not included in the study.

► Those who agreed to be interviewed may differ from those who not agree, thus limiting the generalisability of the findings.

► People with fluctuating cognition may report differing perspectives within the course of a single research interview.

## BACKGROUND

There are over 9.9 million new cases of dementia each year worldwide.[1] Dementia is a progressive condition that describes a set of diseases of the brain, which cause 'memory problems, changes in mood and behaviour, and communication and reasoning problems'.[2] Alzheimer's disease is the most common cause of dementia.[2] In the UK, efforts to raise awareness such as the Prime Minister's national challenge to fight dementia in 2015,[2] have increased public awareness of dementia, and as a result, increasing numbers of people are being diagnosed[3] and this is set to rise further with our ageing population. In 2001 in the UK, the Department of Health recommended that all specialist mental health services for older people include specialist memory clinics.[4] The memory clinic process involves testing, delivery of the diagnosis (usually by the consultant/old age psychiatrist) and agreeing a treatment/care plan. Testing involves reviewing patient history, an interview with

a companion, physical examination, a brain scan and brief cognitive testing[5] using standardised instruments such as the Mini Mental State Examination (MMSE)[6] or the Addenbrooke's Cognitive Examination III (ACE-III).[7] Some memory clinics facilitate a same-day assessment and diagnostic service; however, in most clinics, this takes place over 1 or 2 months. In addition to the memory clinics, there have been recommendations for providing prediagnostic counselling and postdiagnostic support.[5 8] However, patients, carers and professionals express concerns about support before and after the diagnosis.[9 10]

Receiving a diagnosis of dementia is a life-changing event and involves coming to terms with physical and mental deterioration.[11] Due to the nature of the illness, people living with dementia experience comprehension, memory and word finding difficulties, affecting their ability to participate in conversations.[11–13] This in turn leads to changes in competencies (eg, skills, autonomy) and in patients' relationships with others.[11–13]

Since the 1990s there has been an effort to shift the focus of health and social care research from a solely biomedical one, to one that includes peoples' experiences, social factors and the impact of the diagnostic label.[14–17] In dementia, research that includes the lived experiences of people with dementia has grown in recent years[11–13 18–20] including identifying the diverse emotions that people experience when they receive a diagnosis,[12 21–23] and the impact this might have on one's sense of self, autonomy and agency.[12 24–28]

When receiving a chronic and life-limiting illness diagnosis such as dementia, one's identity can be replaced by the available disease identity[29] which carries the stigma attached to this diagnosis.[29–31] In dementia, stigma and marginalisation[27 31 32] add to the difficulties people face in relation to their symptoms. This stigma is widespread, for example, in physicians, patients and their families' perceptions and attitudes.[32] Stigma impacts psychosocial well-being[24] and can affect people prior to being diagnosed, for example, in seeking diagnosis,[33 34] or cause people to feel marginalised after receiving the diagnosis.[35] In line with this, the WHO has emphasised the importance of addressing the stigma attached to dementia,[35] and research has increasingly focused on how people can overcome stigma and preserve their sense of self.[25 26 36 37] A focus on how people can 'live well' with dementia[13 38 39] in a supportive environment has also been highlighted in research[40–43] and policy,[2] that aims to increase the quality of community-based support by the creation of 'dementia-friendly' communities.

Although in recent years there has been growing research on the experiences of people with dementia, this is still limited,[12 13 22] and particularly at the critical period soon after receiving the diagnosis.[12 18 44] An important implication for research at the time of diagnosis is also the fact that many people may have had dementia for some time prior to diagnosis. Hence, the aim of the current study was to explore people's experiences of cognitive decline and receiving a diagnosis of dementia soon after receiving the diagnosis.

## METHODS

Data were collected as part of large mixed-methods study Shared Decision Making in Mild to Moderate dementia (ShareD) conducted across nine memory clinics in the UK including rural (three memory clinics in Devon) and urban areas (six memory clinics in London). Data were collected from May 2014 to July 2016.

Memory clinic staff identified patients with upcoming memory clinic appointments who had a scheduled diagnostic feedback appointment and did not need an interpreter. Information sheets were sent with patient appointment letters. Clinicians assessed whether patients had capacity to consent to participate. When the patient and their companion arrived at the clinic, a researcher approached them to discuss the study further and obtain written, informed consent. For patients without capacity to provide informed consent to participate (one patient in our sample), we followed the 'Guidance on nominating a consultee for research involving adults who lack capacity to consent'.[45] Patients who participated in the ShareD study (n=215; consent rate 51%) and received a diagnosis of dementia were asked at this appointment if they would be willing to undertake an interview about their experiences. If they agreed, a researcher called them in the days after the appointment to arrange a date and time. Semistructured, in-depth Interviews took place in patients' homes within 2 weeks of receiving a diagnosis. All participants gave informed consent. The participants had mild to moderate dementia and their impaired cognitive capacities could lead to misunderstanding their involvement in the study. Their companions were always present when the study was explained to them, and participants were able to opt out at any stage of the research, including the interview. As the interviews involved a discussion of the patient's illness, which may be distressing, it was made clear to them that they did not have to discuss topics that they did not wish to. Interviews were conducted by PX and two other female researchers (ShareD study). All interviewers had PhDs, interview training and research experience.

The interview guide was developed with the Alzheimer's Society (dementia support and research charity) Research Network. Interviews aimed to elicit participants' experiences of receiving a diagnosis of dementia. The interviews were video recorded using GoPro HERO3 cameras. Although the interviews were video recorded due to the equipment available, audio files were transcribed verbatim and transcripts analysed using thematic analysis. Identifying information was removed in order to preserve anonymity. A process of data reduction and display was undertaken using NVIVO and Excel, generating categories, subthemes and themes.[46] This was done in 11 analytic meetings, where the codes were developed, defined and refined. Thematic analysis involved

identifying patterns within the data, by generating codes, categories, subthemes and themes.[47] The analysis was jointly conducted by both authors and discussed in wider analytic meetings increasing the validity of the analytic process. Differences of opinion were resolved through discussion of the empirical data.

## Patient and public involvement

A representative from the Alzheimer's Society, one patient and one carer were on the project management group, which met once a year. In order to ensure maximum clarity for potential participants, the information sheets were designed with the input of the Alzheimer's Society Research Network volunteers (group discussions), who have personal experience of dementia—living with the condition or as a carer or former carer. A workshop with service users, carers and professionals was conducted at the end of the study, where we presented and discussed the study findings. Feedback helped to contextualise/ interpret findings.

## RESULTS

Sixty-one adults who were diagnosed with dementia were interviewed. Interviews lasted on average 22 min (SD=11). Most participants were diagnosed with Alzheimer's disease and were on average 81 years old. Fifty-six per cent were female. Participant characteristics are presented in table 1. Doctors administered either the Mini Mental State Exam (MMSE)[6] or the ACE-III[7] and its shorter version (MiniACE)[48] as part of routine assessment. Sixty patients completed one of these with most scoring in the mild, and some in the moderate, stages of cognitive impairment (table 2).

From 104 codes, 22 categories were identified, which fed into nine subthemes and four overarching themes (see table 3). The four main themes were: (1) accommodating to the diagnosis: threat to self and social identity through increasing visibility of dementia; (2) vulnerability and being in limbo; (3) change in relationships and social isolation: loss of control and agency; (4) maintaining agency and self-worth.

Receiving a diagnosis of dementia was associated with losing competence and agency, thus threatening one's core sense of self. Faced with this loss and the additional stigma of dementia, people struggled to accept the diagnosis: many made considerable efforts to adjust to the news that they are now a person with dementia with others resisting the diagnosis altogether (see figure 1). People were confronted with their decreasing competence in everyday tasks at home, which was becoming increasingly visible in the wider family and with friends and acquaintances in their social networks (see figure 2).

## Accommodating to the diagnosis: threat to self and social identity through increasing visibility of dementia

Peoples' conceptualisations of dementia drew on the explanation of dementia in the memory clinic, personal

### Table 1 Participant characteristics

|  | N | % |
| --- | --- | --- |
| **Site** | | |
| Urban | 11 | 18 |
| Rural | 50 | 82 |
| Age | | |
| Average age | 81 | |
| Range | 65–91 | |
| SD | 6.4 | |
| Gender | | |
| Female | 34 | 56 |
| Male | 27 | 44 |
| Diagnosis* | | |
| Alzheimer's | 36 | 60 |
| Vascular dementia | 12 | 20 |
| Mixed dementia | 10 | 16.4 |
| Lewy body dementia | 1 | 1.6 |
| Parkinson's dementia | 1 | 1.6 |
| Semantic dementia | 1 | 1.6 |
| Education level | | |
| School | 39 | 64 |
| Further education (education below degree level) | 14 | 23 |
| Higher education | 8 | 13 |

*See online supplementary table 1 for descriptions of dementia type.

experience of seeing others (sometimes close family members such as spouses) with dementia, and wider societal depictions of dementia in the media, particularly television and newspapers. Dementia was conceptualised as a brain disease or a natural progression of ageing. It was understood as a continuum with people differentiating themselves from others with more severe symptoms of dementia.

### Differing perspectives on dementia as an illness versus normal ageing

People were faced with the information that they had dementia, a brain disease that was different to and distinct from normal ageing. Dementia was characterised primarily as pertaining to memory loss due to an illness (often shrinking) of the brain: 'my brain is shrinking' (P102) (Indicates participant number), 'memory loss, and I'm not going to be able, you can't bring it back' (P07), 'this short term memory loss is the problem' (P115). This was also used to explain dementia in other people: 'she can't talk much, she can't walk, it's like come on! Because her brain's unmotivated' (P182).

Many described dementia as a natural part of ageing: 'I know that it affects elderly people…Develops as you get older' (P161), that affects everyone: 'It happens to

**Table 2** Cognitive test scores

| Cognitive test* | N | Minimum | Maximum | Mean | SD |
|---|---|---|---|---|---|
| Score on MMSE (out of 30) | 5 | 16 | 28 | 21 | 4.8 |
| Score on ACE-III (out of 100) | 53 | 47 | 94 | 71 | 11.6 |
| Score on MiniACE (out of 30) | 2 | 8 | 10 | 9 | 1.4 |
| Total | 60 | | | | |

*See online supplementary table 2 for description of tests.
ACE-III, Addenbrooke's Cognitive Examination III.

everybody in the end. It's where we're all going to go as you live longer' (P102). Dementia was also described as loss of competence: 'a stage where perhaps you're not capable of carrying on your own life without assistance from others' (P117). They described other people with dementia as being dependent and vulnerable: 'My mother had Alzheimer's…We were told to get her a silver identity bracelet, but not put her name on it, put her phone number' (P57), and regressing to a childlike state: 'I thought dementia was like a complete memory loss and you went back to you're a childlike' (P203).

A few people (n=5) stated that they did not know what dementia was.

### Personal experience of and distancing oneself from others with more severe dementia

There were varying levels of exposure to dementia in others, usually other family members or friends, and some compared their 'stage' of dementia to that: 'I don't think it is that bad is it, well you see people with lots of memory problems so you know what degree' (P117), and differentiating themselves from others with more severe symptoms: 'I have a lot of friends who are a lot worse than me' (P118). In this regard, dementia was described as a continuum from less severe to very severe: 'I've realised that there's several degrees of it' (P200).

For some people expectations and reactions were influenced by images of dementia in the media:

you listen to the television and the radio and things like that and you know you see things that suit the way

you're thinking you know and you hear things that you think like 'oh I hope I ain't got that' you know (P211).

### Affective response to the news: from shock and terrified to less affected

Reactions ranged from expecting the diagnosis and acceptance to being surprised and shocked. Over a third of people (21) accepted the diagnosis while around half (29) found it difficult to accommodate the news. Many expressed surprise as their own assessment of their symptoms differed from the memory clinic outcome: 'I was quite keen to go. But this is because I think 'well my memory's not so bad' so I was looking forward to them telling me 'oh your memory's alright' (P117). Sadness and fear was a common reaction: 'it's is really frightening' (P86), as was shock at the news: 'It was a shock. A sad shock', and disbelief: 'I didn't know anything like that would happen to me' (P102). There was struggle to process the news: 'it has happened… I've learnt to live with it but I can't accept it yet' (P07). On the other hand, some stated that they were aware of their memory decline: 'I was perfectly prepared to be told this…I knew it wasn't my mind playing tricks…I was getting worse' (P07), and described efforts to remain positive: 'I try not to dwell on it really' (P184).

A third of people did not acknowledge there was a problem with their memory: 'They said about my memory and I thought 'what are they on about?' There's nothing

**Table 3** Themes and subthemes

| Themes | Subthemes |
|---|---|
| Accommodating to the diagnosis: threat to self and social identity through increasing visibility of dementia | ► Differing perspectives on dementia as an illness versus normal ageing.<br>► Personal experience of and distancing oneself from others with more severe dementia.<br>► Affective response to the news: from shock and terrified to less affected.<br>► Visibility to others, stigma and fear of disclosure. |
| Vulnerability and being in limbo | ► Confronting an inability to do 'normal' things and socialise.<br>► Being in limbo—future disintegration is scary. |
| Change in relationships and social isolation: loss of control and agency | ► Loneliness and social exclusion.<br>► Increasing burden and dependence. |
| Maintaining agency and self-worth | ► Adapting and staying positive.<br>► Promoting agency and independence. |

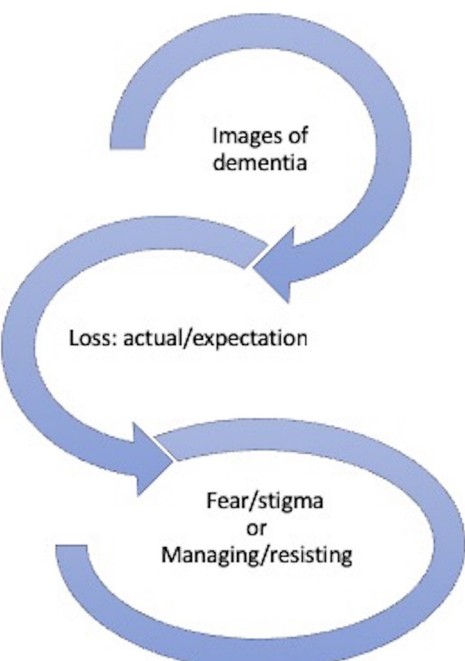

**Figure 1** Impact of dementia images/diagnostic label on the adjustment to diagnosis.

wrong with my memory at all' (P150), or did not accept the diagnosis: 'Well I'm not convinced I've got it' (P193).

However, peoples' accounts were not always consistent, with differing views expressed during the course of the interview. For example, some people who said they were not surprised by the diagnosis: 'I don't think it was a surprise' (P117), they also stated in the interview that they did not expect it: 'I hadn't really thought I had a problem with my memory' (P117).

### Visibility to others, stigma and fear of disclosure

Many people discussed dementia as a progressive loss of mental competence, culminating in being an idiot, crazy, crackers and losing the plot: 'it's, what would you call

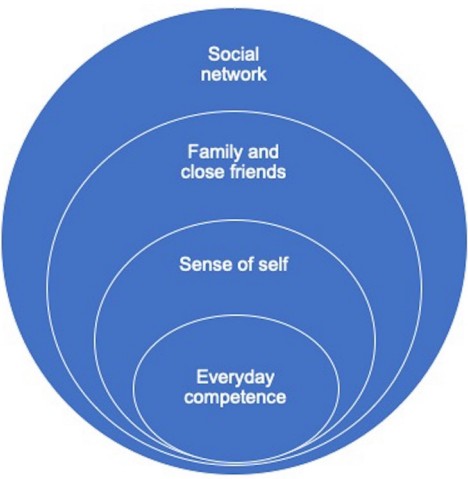

**Figure 2** Impact of close/wider social networks on sense of self and competence.

it, defamatory, derogatory? It shows you're going a bit crazy I suppose, It's slightly embarrassing' (P183) with the person with dementia perceived as confused and vulnerable: 'It's a word I associate with being, not being mentally completely competent' (P167). People also drew on their personal experiences with friends/family: 'we've had odd women... this country club where I play, lots of old women who got worse and worse' (P186), and losing one's sense of reality: 'she's lost the plot. Completely' (P174), 'She's just gone, lost it, lost it' (P29). People were afraid of deteriorating in this way: 'I don't want to become an idiot you know' (P86). They worried about becoming angry: 'it worries me that I'll lose my temper. Because you know that I worry that I'm going to do somebody some harm' (P07). There was an expectation that changes in their personality might occur, e.g. might become dangerous, 'if I was a danger I should go into a home' (P86).

For some people, the dementia label was problematic and euphemisms were used instead: 'you're using dementia. We were using perhaps a word more like forgetfulness not dementia' (P186) and did not identify with having 'mental' problems: 'he (doctor) said that I've got problems mentally...but for me I really feel I've just got loss of memory' (P46).

Many discussed worries about being seen to lose competence by family, friends and acquaintances. They were aware that their symptoms were or would become increasingly visible to others: 'I keep worrying you see in case I'm doing something...how quickly will I get very noticeable?' (P172). Some felt they had to put additional efforts into hiding their symptoms: 'he said the children had noticed...I didn't ask which children or which member or why. And I thought oh dear I must pull my socks up' (P186), and having to find alternative explanations for their dementia symptoms:

> I have to cover up, you know. Well, say I'm in a conversation with somebody and I have to admit to them either that I have lost the plot completely or something disturbed me or something. I have to find an excuse for not getting it right (P84).

People reported not telling others, even close family such as children about their diagnosis because of the stigma of dementia: 'I said to her, don't you say about that because you know what the village is like' (P149), 'I said to (husband's name) 'I don't want people to know I'm going crackers' (P204), or only disclosing to close family members. There was an expectation that others' behaviour towards them would change: 'you don't know how people will be. I haven't told my daughter yet. I don't know whether to or not you see because they start to treat you differently don't they?' (P172). A participant also expressed worry of becoming discredited if others found out:

> so quite genuinely I might forget to do something which everybody does ... so I don't want them saying

'well of course she's…' so I decided I didn't want to tell anybody (P117).

## Vulnerability and being in limbo

People were concerned about the current impact of dementia on their lives which left them feeling frustrated and vulnerable. This was compounded by a feeling of being 'in limbo': while the changes that had already happened were challenging to come to terms with, people were profoundly affected by not knowing how much they would deteriorate and when this would happen in the future.

### Confronting an inability to do 'normal' things and socialise

People were confronted by an inability to do normal things that used to be easy for them. This made them frustrated and irritated: 'I find that I've got to ask somebody else the date or whatever that vexes me a bit' (P120): 'I can't remember the ingredients of dishes that I made for years and years. And that makes me very frustrated, because they're ones, like, that my son loved' (P84). Dealing with financial matters was challenging: 'things like PIN numbers and banking and that sort of frustrating. You feel very vulnerable for a bit' (P199).

Symptoms prevented people from carrying on with usual activities and hobbies: 'normally I go bowling at (place), that's on Wednesday nights, but I haven't been much lately… I haven't been quite right' (P104). Many recognised that it was not safe to drive anymore: 'I don't like driving children and when a little bit iffy. It wouldn't be good… Especially if a doctor said don't drive' (P205). Driving was described as an important aspect of people's life and independence which was no longer available: 'I enjoyed driving…if I was feeling sort of lonely or anything' *(P86)*. For many this impacted on keeping in contact with others: 'even the walk to the post box on (street) is over a mile away' (P187) and being able to maintain intimate relationships: 'the trouble is, I've got a wife far away here, I've got to go there. It's not easy' (P182).

### Being in limbo: future disintegration is scary

People were attempting to come to terms with future decline: 'it's going to get worse nd that is the most worrying thing' (P207), 'because there is going to come a time when I won't remember my name' (P07). They recognised that this would also be emotionally difficult and lead to a loss of self-worth: 'would make me feel… probably upset and useless later on' (P187). They discussed the fact that there is no cure for dementia: 'he did say there is no cure for this' (P117), and that the medication could delay deterioration: 'the doctor was very honest, and said like think that the best that can happen is that I don't get any worse, they can't see it getting any better' (P133).

People talked about the uncertain trajectory of the disease: 'it's a bit like being in limbo, a bit you know, not really knowing what's going to happen' (P19). There was uncertainty of the kinds of changes and decline they should expect: 'will it get worse or what, or will it just take

it's time and I'll just be forgetful' (P145) and hoped that they would still be able to retain some capacity: 'I just hope I'm capable… to a degree to be alert' (P36).

## Changing relationships and social isolation: loss of control and agency

People experienced changing dynamics in close relationships and wider social networks. Loneliness, increasing dependence on others and feelings of becoming a burden on others led to loss of control of one's own life and loss of personal agency.

### Loneliness and social exclusion

People expressed concerns about being socially isolated: 'I'm by myself. Which makes me a bit lonely. And I spend a lot of my life watching television' (P205), and reduced opportunities to meet others: 'I don't meet a lot of people anyway not now' (P207). Withdrawal from social activities was noted:

> I haven't played actually (bridge). (Spouse) was only saying at lunch time do you realise it's a long time since you played. And then I thought is that because I am forgetting that I've been sort of excluded without thinking much about it (P186).

Symptoms also had an impact on close relationships at home: 'I noticed that I've become more quiet even with (spouse) at meals, you know, too. And she takes to do puzzles instead of communicating' (P83).

People anticipated having to leave their home. This was very unsettling and would lead to further social isolation: 'I worry that it'll get worse and if it does and if I become a nuisance, I'll be moved on somewhere else. Because I've seen that happen here' (P84). People expressed a loss of choice regarding their living arrangements: 'you're not locking me up, it don't matter how bad I get' (P207). Being uprooted from one's home and community would lead to further social isolation:

> they said they would if I couldn't cope here, they would get a place in (place), and of course (daughter) goes on and on. But I hate (place). There's nothing there! And I've got friends here, and we go and have a cup of coffee (P57).

### Increasing burden and dependence

People expressed concerns and uncertainty about their relationships with others: 'I'm not sure how that you know, relationships, you know, what comes from that' (P19). For others, dementia had already changed the dynamic with family members:

> She's also very fast compared to me. Sometimes too fast. Oh, she must get fed up with my not understanding or not being… she finds it difficult to see my lack of understanding or knowing what's going on (P83).

People expected that the diagnosis would have ramifications regarding how they are treated by others: 'they'd

probably lock me up and throw away the key. I don't know, my daughter sort of I don't think she got much patience' (P07). There were concerns about becoming a burden: 'I feel 'oh poor (name), he's got to do this and that' you know? Because in a way he's my main carer' (P19), and described efforts to reduce the burden: 'I'm willing to help and do what I can and not be a nuisance to everybody else' (P46). People who lived on their own anticipated their future reliance on others:

I wouldn't like to be living on my own with it I must say…but people have their own lives as well don't they…I suppose that's a bit of a sort of worry, you know, if you become, you know, more dependent on other people (P19).

Handing over decisional responsibility was also expected: 'well I'd have to let her do it wouldn't I?…I wouldn't really have a lot of choice would I?' (P29). Losing agency was difficult for people and half of the participants expressed a preference for exercising agency: 'I don't think anybody else should be involved in it except myself. I don't want any of my so-called family to be making decisions for me' (P86).

### Maintaining agency and self-worth

Around a third of people expressed a positive outlook, by focusing on what they could do and the benefits of receiving a diagnosis. This involved acknowledging dementia and an explicit decision to accept the diagnosis and adapt to changes. They reported using coping strategies to help them to deal with memory problems and accepting support from others. Adopting this perspective helped people to maintain a sense of agency and a valued sense of self.

#### Adapting and staying positive

In the face of a diagnosis of dementia, some people expressed a positive outlook: 'I'm happy in myself, you know, it's just that I keep forgetting things' (P145). They focused on what they can do: 'I'm just very thankful I can go out, I've been out for the paper and things this morning, I can do things like that' (P118). Others talked about the usefulness of having a diagnosis: 'having a diagnosis, it's also helpful you know, it's not quite so totally unknown and I don't know what's the matter with me' (P19).

Another way to stay positive for people was to validate competence by demonstrating competence. It was common that people focused on the encouraging aspects of their diagnosis meeting: 'he was talking about old people get aggressive and stuff like that and he just pointed at me and says I can't see you being like that. So it was a compliment to me' (P36).

People also provided alternative accounts to downplay their memory problems, for example, not being academic: 'I'm a guy that I work with my hands all my life, I'm not a pen pusher, never been a pen pusher'. *They* focused on their past abilities and used their life stories

to minimise symptoms and promote a capable self: 'I had plenty of brain there when I was putting car engines together' *(P182)*. When accounts of others about the oneself were inconsistent with their sense of self, people offered alternative accounts:

he said 'Mum seemed to have been a bit slow in catching up with what we were talking about', well I was probably reading something else at the time, you know? I wasn't really interested in what they were talking about (P149).

Attributing forgetfulness to normal ageing helped to normalise dementia: 'I mean I can forget things but mainly it's just old age' (P131). This way the dementia diagnosis was accepted as part of normal ageing: 'when you're 90, you know, you're not answerable for all your bits and pieces' (P138).

#### Promoting agency and independence

Adopting coping strategies and accepting practical and psychological support from others could promote a sense of agency and independence. People described techniques they used to actively adjust to symptoms: 'I can read a book… what I do is have little kind of post cards and I write the names of the characters or dates and then if that one connects to that one I do a connecting thing' (P156). Keeping a diary was a common strategy to deal with memory problems: 'I started taking a pad in my bag and writing everything down that I thought I ought to remember' (P186).

Support from family, friends and the community was fundamental in more and less subtle ways, from family coming to regularly check on the person to seemingly small but important exchanges that helped people to maintain independence: 'I am happy and comfortable, because I have, I'm lucky enough to have daughters to keep an eye on me' (P101), 'the butcher and the baker and the veg man, they greet me by name…I can point to what I want and they get it and then I give them my purse… and they do the money' (P156). Family often acted as an advocate for the person with dementia: 'because I can't remember everything…and if someone's talking to you they tell you something, but then you might start thinking about what the… But, they're still talking…So, it's better two of you' (P53). People coped by asking others to verify things for them: 'If I had a lot of instructions to do… probably tomorrow I'd probably say to (name) 'hey is this what I've got to do today?' You know. Check it through' (P172).

Some people planned for the future: 'I know I've done everything I can do and everything is paid for and organised, you know. So really he has nothing to worry about' (P84). Maintaining positive relationships was also mentioned: 'we journey along quite happily. We haven't let it worry us very much' (P186). For many people dependence on others was not presented as negative: 'we live together (with daughter), so there's going to be ups

and downs. I accept that' (P07), seeing reliance on other people as support rather than loss.

## DISCUSSION

This study captured peoples' initial subjective experiences of receiving a diagnosis of dementia. People were interviewed with 2 weeks of receiving their diagnosis. People expressed a range of understandings and reactions to the diagnosis. Over a third of people did not expect this diagnosis and were shocked, and a third of people were not surprised and/or less affected. However, many people had a fluid state of emotions, involving acceptance and rejection of the diagnosis. This is in line with the 'paradox' of dementia expressed through coexisting views of acknowledgement and resistance,[23 49] where people struggle to maintain a valued sense of self against the loss and stigma attached to this diagnosis.[21 24] The experiences reported in this paper should be understood as products of the contingencies of the interview situation,[50] that is, findings are situational and reflect accounts and emotions at the time of the interview which may change over time. Peoples' experiences were sometimes fluid within the interview, reflecting how an interview somewhat artificially breaks down these experiences into component parts.

People described loss on multiple levels[12 21–23]: a loss of control over their lives due to loss of competence in everyday normal activities of daily living (eg, driving, managing finances, not remembering how to cook, not remembering what they were supposed to be doing on a particular day) and this foreshadowed further future loss, of mental competence and of being able to live or be cared for in their own home. They were aware that the decision about being uprooted from their familiar surroundings would most likely not be theirs to take. They were perturbed by the prospect of moving out of their home and community, reflecting peoples' basic needs for a familiar place of shelter[51] and maintaining close relationships and friendships. Similarly, people were worried about the impact of cognitive impairment on driving (eg, losing the ability to drive/driving licence). This impacted on peoples' feelings of security and identity.

Over 30 years after Charmaz's[27] seminal work, people invoke the stigma of dementia that brings social isolation and exclusion,[52] restricts what they can do, makes them feel discredited and a burden on others. People understood dementia as a regressive process of becoming child-like[53] and becoming crazy, resulting in efforts to hide their diagnosis from friends or family to avoid embarrassment.[54] Expectation that one's identity will change in the eyes of others,[55] prevented people from gaining the benefits of sharing their diagnosis, which include improved quality of life, better planning for the future and developing a positive support system that promotes independence.[54 56] While recent efforts to explore and identify stigmatising and discriminating discourses, by including people's subjective experiences in research have been

successful,[57] these need to translate into practical solutions that will help clinicians and families to better deal with these issues.

These experiences of loss are so significant that efforts and coping mechanisms to restore a sense of self and independence are very important. We found that people developed a range of strategies to maintain a valued self and their independence,[39 54] by normalising, for example, by describing it as 'normal ageing',[58] resisting or downplaying the seriousness of the diagnosis. This also demonstrates that for some people, symptoms are disregarded as normal ageing prior to receiving the formal diagnosis, and therefore the point of diagnosis does not always closely correlate to the point at which dementia symptoms commence. Other people incorporated their diagnosis into their existing self-concept,[55 59] maintaining a continuous sense of self[54 60] by drawing on abilities and past successes that were presented as still part of their identity. People who focused on maintaining a positive outlook viewed dependency as support rather than a burden on others. The complex issues, for example, uncertainty about the future and stigma, can be discussed with patients and their family/carers early, that is, at the diagnostic feedback meeting or soon after, as we have previously found that patients/carers express these concerns during the diagnostic feedback meeting.[61] This is crucial for the diagnostic meeting agenda as appointments in memory clinics typically lasting 30 min and post-diagnostic support not being available for all patients.

In line with previous research, this study found popular dementia images, social interaction and maintaining positive relationships to be central in shaping self-perception, well-being and independence.[25 26 40 41 43] We found that broad relationships, for example, activity groups and communities were important for the person's identity and maintaining a valued self. This was often done in subtle ways (eg, butcher helping with counting money) that can maintain peoples' independence. The findings also lend support to psychosocial theories of self-concept and illness[62–67] that emphasise the importance of social representations and social interactions in sustaining perceptions of self. Thus, demonstrating a link between the other peoples' reactions towards the person with dementia, to the experience of symptoms and adjustment to dementia.[64 65 67]

In this context, the development of 'dementia friendly communities' can have a positive impact as it aims to improve the physical environment and quality of life of people living with dementia.[67–70] Similarly, the quality of relationships with family and friends shaped people's positions in terms of independence, agency and feeling secure. This is particularly important as many people with dementia in the UK live with or near family members.[43]

Receiving a diagnosis of dementia is a communicative *process*. Peoples' subjective experience of cognitive decline prior to receiving a diagnosis is relevant to this process. Adding to this, are expectations of how others might react to the diagnostic label, for example,

becoming stigmatised/excluded. In addition, we saw how dementia differs from other progressive diseases as people experience 'cognitive fluctuations' (interruption in the ongoing flow of awareness).[71] These factors influence how the diagnosis is received and accommodated, and has implications for the diagnostic process, including prediagnostic counselling, and postdiagnostic support. Support prior to receiving the diagnosis would not only make the process less confusing for the patient[72] but also help people who care for them, as due to the nature of the illness, it is typically carers who initiate assessments for possible dementia.[22] It would also *prepare* people for this diagnosis, as we found that over a third of the participants did not expect the diagnosis and many did not know what dementia is. However, only one of the nine participating memory clinics, had a formal process in place for prediagnostic counselling, despite recommendations that this should be part of the diagnostic process.[8] In addition, despite the benefits of postdiagnostic support for adjusting to the diagnosis and enabling patients and their families to plan for the future and increase independence,[73] there is still lack of adequate postdiagnostic support.[10 74]

## CONCLUSIONS AND FURTHER RESEARCH

Receiving a diagnosis of dementia is often a turning point in peoples' lives. People are confronted with and frustrated by decreasing competence in everyday normal activities. They feel vulnerable and in limbo not knowing what further deterioration they will experience and when this will happen. People are worried about disclosing their diagnosis to others, with some actively concealing their diagnosis even from close family. Limited time in diagnostic appointments and limited prediagnostic counselling and postdiagnostic support means people have few opportunities to address the emotional impact of being told one has dementia. Future research could address how to optimise participation of people with dementia in diagnostic feedback meetings, for example, through spending more time orientating people to the purpose of the meeting and addressing the emotional impact of a diagnosis of dementia. The stigma experienced by patients could be explicitly addressed to mitigate the threat to one's self-identity and social identity. Finally, professionals could discuss the benefits of staying positive and helpful coping strategies used by others along with accepting support to live well with dementia.

Future research could address how to tailor the diagnostic process to people with fluctuating cognition, for example, reorientating people to the purpose of meetings during the diagnostic process, addressing the emotional impact and stigma of a life-changing diagnosis such as dementia and providing hope by focusing on how to live well with dementia. Further research can also explore the relationship between participants' sociodemographic factors (eg, age, education) and attitudes/concerns regarding cognitive decline and receiving the diagnosis.

**Acknowledgements** We thank all the patients who participated in the study, as well as the ShareD project team. We would also like to acknowledge the support of the National Institute for Health Research, through the Comprehensive Clinical Research Network.

**Contributors** RM conceived the idea for this study and obtained project funding. PX contributed to the data collection. RM and PX were responsible for data analysis and interpretation of data. PX prepared the first draft of the paper. RM and PX contributed to drafting of the paper and approved the final manuscript.

**Funding** This study was funded by the NIHR Research for Patient Benefit (RfPB) Programme (Grant Reference Number PB-PG-1111-26063).

**Competing interests** None declared.

**Patient consent for publication** Not required.

**Ethics approval** The study obtained ethical approval from the Camden and Islington Research Ethics Committee (13/LO/1309).

**Provenance and peer review** Not commissioned; externally peer reviewed.

**Data sharing statement** Interviews and observational data from this study are personally identifiable for patients and staff, and are therefore not available for data sharing.

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
