## [Reviewer comments · BMJ Open]

ARTICLE DETAILS

TITLE (PROVISIONAL)	Subjective experiences of cognitive decline and receiving a diagnosis of dementia: Qualitative interviews with people recently diagnosed in memory clinics in the United Kingdom.
AUTHORS	Xanthopoulou, Penny; McCabe, Rose

VERSION 1 – REVIEW

REVIEWER	Mark Schweda Department of Medical Ethics and History of Medicine, University Medical Center Göttingen (Germany)
REVIEW RETURNED	19-Sep-2018

GENERAL COMMENTS	This is a very interesting paper dealing with an important and current topic: patients' experience of being diagnosed with dementia. Based on a thematic analysis of 61 semi-structured interviews with people recently diagnosed with dementia in 9 UK memory clinics, the authors describe the subjects' experiences of and reactions to the diagnosis and draw conclusions for providing information and consultation in the context of dementia diagnosis. The text is very well written and argued and it provides illuminating empirical insights into patients' subjective experience and concerns. I only have a few recommendations for sharpening the focus and further strengthening the argument: - Title and focus: I think the title and the research interests stated in the background section do not fully reflect the study's actual scope and implications. As a matter of fact, the analysis not only focuses on patients' experience of being diagnosed with dementia, but rather on their experience of cognitive decline and the ways the diagnosis itself plays into that experience. Thus, many findings do not specifically pertain to the diagnosis but rather to general issues of being confronted with "an inability to do normal things and socialise" (2.1), "loneliness" (3.1), or "increasing dependence" (3.2). This broader focus is important since it may provide information on people's possible motivations for seeking diagnosis in the first place, as well as on the interplay (and sometimes discrepancy) between personal experience and clinical diagnosis. However, it would be important to differentiate more consequentially between the general aspects of experiencing cognitive decline (or not) and the specific effects of the diagnostic process and label itself. This would also allow for discussing processes of pathologisation, medicalisation, stigmatisation and discrimination in a more detailed way (see for example the study by Beard and Fox in SS&M 2008). - National context: For an international audience, it would be helpful to learn a bit more about the framework conditions and practical
---

	processes of dementia diagnosis in the UK: Is it usually conducted at memory clinics? What professions are involved? What are the standard diagnostic methods employed (neuropsychological, biomarkers)? How does the process usually work? What do the relevant guidelines or standards say about advance patient information, disclosure of diagnosis and counselling? This could also be important for formulating practical recommendations in the conclusions section. - Methods: The authors state the interviews were video-recorded. What was the rationale behind this methodological decision? Was the video material also analysed and, if so: how? What does this analysis contribute to the interpretation of the findings? - Research ethics: The research took place within two weeks of learning about the diagnosis. One would assume that this can be a particularly difficult time for the patients. What was done to avoid imposing any additional psychological distress through the interviews? How did the subjects themselves evaluate the experience of participating in the study? - Socio-demographic factors: Although this is a qualitative research design, it would be interesting to learn more about the role of socio-demographic factors like age, gender and education. Can the authors make any statements / formulate hypotheses about the influence of these factors on participants' attitudes and concerns regarding cognitive decline and dementia diagnosis? - Role of media images: In their analysis, the authors point to the influence of social depictions of dementia in the media (p.6). Could you elaborate a bit more on this? What kinds of media images were mentioned and what role did they play for participants' experiences and concerns? - Discussion: The authors point out that their own findings are in line with previous studies. I would recommend they also highlight how their own research maybe differs from, further contributes to or even points beyond the current state of the art: Did unexpected aspects emerge? What new questions came up? For example, I thought it was quite interesting that over a third of the participants actually did not seem to expect the diagnosis or even rejected it (and some did not even know what dementia is). What does this mean? What does it say about the diagnostic process and label and its communication? - Limitations: The discussion section should include a para providing a critical reflection of the study's methodological limitations and their possible impact on the outcomes and their interpretation. It also wouldn't hurt to point out open questions and need for further research.
--	---

REVIEWER	Melissa J Bloomer Deakin University, Australia
REVIEW RETURNED	19-Oct-2018

GENERAL COMMENTS	Thank you for the opportunity to review this manuscript submitted to BMC Open. Whilst a very worthy topic, the paper itself could do with some refinement. Please also remember that the journal has an
---

	international readership so the manuscript must be written in a way that it would be easily understood by the international audience. In the INTRODUCTION, there are a number of statements that I would suggest should be supported by an academic reference. These include: '...increasing numbers of people are being diagnosed and this is set to rise further with our aging population'. 'Receiving a diagnosis of dementia.... physical and mental deterioration' 'Due to the nature of the illness....participate in conversations' It is also unclear what is meant by 'changes in competencies' In the second paragraph, the first sentence commencing with 'Traditionally, research has ignored...' is a very broad and sweeping statement and I do not agree. Patient experience is fundamental to healthcare and there is a substantial body research work around this. There may be less in relation to people with dementia, but I think it is incorrect to state that research has ignored patients' experiences. In relation to the experiences of people with dementia, you may wish to look at the work of Jan Dewing and more recently Robin Digby. In the third paragraph, I think it is important to state that not only is dementia a chronic illness, but it is also a life-limiting illness. Also, it is unclear what is meant by the sentence commencing with 'The stigma is widespread....' – please make this clearer. Whilst I understand that the aim was to explore peoples' experiences soon after diagnosis, the obvious issue here is that the point of diagnosis does not always closely correlate to the point at which the dementia and its symptoms commenced. Some people may have symptoms often disregarded as 'ageing' for some time prior to receiving the formal diagnosis. I think this would be an important point to note, particularly since your results reveal some patients scored as having moderate cognitive impairment at the time of their interview. You refer to 'small sample sizes at the bottom of page 3, as if it is a limitation. I would suggest that it is not, but rather the sample sizes are more indicative of the human nature of this research. If you wish to critique sample size then this should be done in relation to other factors such as the transferability of the results and data saturation, not the sample size itself. Page 4, for the purpose of clarity, please describe what is meant by a 'Memory Clinic' and how it is used in the UK setting. At a clinic appointment for 'scheduled diagnostic feedback', does that mean individuals are given the news that they have dementia? if so, I would question whether it was appropriate to attempt recruitment for this study at the same time. How was 'capacity to consent' assessed by clinicians in the memory clinic? Was consent to participate in the study reaffirmed immediately prior to commencement of the interview? Were participants allowed to have support persons, such as their spouse present at the time of the interview, and if so, how was their presence managed? Was a list of questions or guiding prompts used to guide the interviews? Were they structured, semi-structured or unstructured? There is also growing evidence for the use of a conversational approach to interviewing people with dementia as it is less confronting and preserves dignity. In the RESULTS section, please also report SD for average age. Can you please explain why you had 61 participants but only 6 completed the cognitive test? Is the cognitive test part of this study specifically, or part of the larger study? If it is not part of this study, then do not report that here – suggest you refer to it in the Inclusion
--	--

	criteria. In Table 1, not sure what is meant by 'further education'? In Table 2 you refer to MMSE, ACE-III and MiniACE. Each of these needs to be explained in the body of the paper. But as suggested previously, if these were used as a criteria for inclusion, then they should be detailed there, not in the results. It is difficult to comment on the findings when it is not clear what the questions were or how interviews were conducted. The conclusion is a bit brief. There is scope to make this more punchy, and clearly articulate the implications and 'so what' of this work. It is important work, but I am not sure you have capitalised on that here. Suggest you also make sure that the conclusion also articulates the future directions for research and how clinical practices associate with dementia care should change as a result I look forward to seeing this published, but think there is some room for improvement in this paper
--	--

REVIEWER	Frans Verhey Maastricht University, Alzheimer Centrum Limburg, Netherlands
REVIEW RETURNED	25-Oct-2018

GENERAL COMMENTS	This is a qualitative study into patients' experiences of being diagnosed with dementia. The results are interesting and very useful; I have read the manuscript with great pleasure. The method seems sound, but I do not consider myself as a specialist of qualitative studies. I have a few remarks that may improve the paper: The paper may benefit from considering the diagnostic disclosure more as a interactive process between the clinicians and does for instance not describe the way the diagnosis was actually being disclosed, e.g., as bad news, an illness, as a handicap for which support is needed etc. I presume this may also have an impact on patients' experiences? In line with this, it may be useful to know more about the types of participating memory services. For instance, how many of them were academic, what were the disciplines of the clinicians who disclosed the diagnosis (psychiatrists, psychologists, neurologists)? The conclusion of the paper that receiving a diagnosis is a confronting turning point in peoples lives seems to be too strong in the light of the findings that about one third of the participants were not surprised and/or less affected. The authors touch upon the loss of the abilities to drive. This is a complicated topic, and bringing this up in the disclosure visit may have practical consequences, which directly impact patients' experiences. I wondered if the authors could elaborate a bit more on this. The first theme ("dissonance") seems to be a bit heterogeneous, as it encompasses quite some contrasting themes (illness vs normal aging, terrified vs. less affected). This seems to be a bit ambiguous, which seems to be somewhat unusual. Perhaps the authors can explain.
---

VERSION 1 – AUTHOR RESPONSE

Reviewer: 1

1. Title and focus: I think the title and the research interests stated in the background section do not fully reflect the study's actual scope and implications... their experience of cognitive decline and the ways the diagnosis itself plays into that experience...
 - The article title has changed to include peoples' experiences of cognitive decline as follows: "Subjective experiences of cognitive decline and receiving a diagnosis of dementia: Qualitative interviews with people recently diagnosed in memory clinics in the UK"
 - The aim of the study has changed from: to explore peoples' subjective experiences of receiving a diagnosis of dementia within two weeks of the diagnosis.

to: explore people's experiences of cognitive decline and receiving a diagnosis of dementia.

2. This broader focus is important ... However, it would be important to differentiate more consequently between the general aspects of experiencing cognitive decline (or not) and the specific effects of the diagnostic process and label itself... This would also allow for discussing processes of pathologisation, medicalisation, stigmatisation and discrimination in a more detailed way (see for example the study by Beard and Fox in SS&M 2008).

Thank you for this suggestion. Text added where we discuss the impact of stigma (p.16): While recent efforts to explore and identify stigmatising and discriminating discourses, by including people's subjective experiences in research have been successful [Beard & Fox 2008], these need to translate into practical solutions that will help clinicians and families to better deal with these issues.

And: The authors point out that their own findings are in line with previous studies. I would recommend they also highlight how their own research maybe differs ... I thought it was quite interesting that over a third of the participants actually did not seem to expect the diagnosis or even rejected it (and some did not even know what dementia is). What does this mean? What does it say about the diagnostic process and label and its communication?

The following text has been added to the discussion: Support prior to receiving the diagnosis would not only make the process less confusing for the patient [Samsi et al., 2014], but also help people who care for them, as due to the nature of the illness, it is typically carers who initiate assessments for possible dementia [Bunn, 14]. It would also prepare people for this diagnosis, as we found that over a third of the participants did not expect the diagnosis and many did not know what dementia is.

3. National context: For an international audience, it would be helpful to learn a bit more about the framework conditions and practical processes of dementia diagnosis in the UK....AND REVIEWER 2 : ..for the purpose of clarity, please describe what is meant by a 'Memory Clinic' and how it is used in the UK settings AND: types of participating memory services. For instance, how many of them were academic, what were the disciplines of the clinicians who disclosed the diagnosis (psychiatrists, psychologists, neurologists)?

New paragraph has been added in the introduction (p.3) explaining the memory clinic process is in the UK:

In 2001 in the UK, the Department of Health recommended that all specialist mental health services for older people include specialist memory clinics.[DoH, 2001] The memory clinic process involves testing, delivery of the diagnosis (usually by the consultant/old age psychiatrist), and agreeing a treatment/care plan. Testing involves reviewing patient history, an interview with a companion, physical examination, a brain scan and brief cognitive testing.[NICE, 2007] In some areas in the UK

(e.g. London) assessments usually take place over one or two months, whereas in other areas (e.g. Devon) assessments (including brain scan) and diagnosis feedback all take place on the same day.] In addition to the memory clinics, there have been recommendations for providing pre-diagnostic counselling and post-diagnostic support.[Hodge et al., 2014, NICE 2007]

4. Methods: The authors state the interviews were video-recorded. What was the rationale behind this methodological decision? Was the video material also analysed and, if so: how? What does this analysis contribute to the interpretation of the findings?

Although the interviews were recorded using video cameras, only audio file transcripts were used for the analysis. This has been clarified as follows: "Although the interviews were video-recorded due to the equipment available, audio files were transcribed verbatim and transcripts analysed using thematic analysis"

5. Research ethics: ... What was done to avoid imposing any additional psychological distress through the interviews? How did the subjects themselves evaluate the experience of participating in the study?

New text added in the methods to explain steps taken during consent and interviewing (p.5):

The participants had mild to moderate dementia and their impaired cognitive capacities could lead to misunderstanding the study. Their companions were always present when the study was explained to them, and participants were able to opt out at any stage of the research, including the interview. As the interviews involved a discussion of the patient's illness, which may be distressing, it was made clear to them that they did not have to discuss topics that they didn't wish to.

6. Socio-demographic factors: Although this is a qualitative research design, it would be interesting to learn more about the role of socio-demographic factors like age, gender and education. Can the authors make any statements / formulate hypotheses about the influence of these factors on participants' attitudes and concerns regarding cognitive decline and dementia diagnosis?

This is an interesting suggestion, however, we did not perform any additional analysis on the relationship between the participant's socio-demographic factors and the qualitative findings. We have added this in the "conclusion and further research" section (p.16): Further research can also explore the relationship between participants' socio-demographic factors (e.g. age, education) and attitudes/concerns regarding cognitive decline and receiving the diagnosis.

7. Role of media images: ... What kinds of media images were mentioned and what role did they play for participants' experiences and concerns?

We have added an example in the results section (page 9): For some people expectations and reactions were influenced by images of dementia in the media: "you listen to the television and the radio and things like that and you know you see things that suit the way you're thinking you know and you hear things that you think like 'oh I hope I ain't got that' you know" (P211)

8. Limitations: The discussion section should include a para providing a critical reflection of the study's methodological limitations and their possible impact on the outcomes and their interpretation. It also wouldn't hurt to point out open questions and need for further research.

We have expanded the limitations in the "Strengths and limitations of this study" section (p.3). by adding an additional point: People with fluctuating cognition may report differing perspectives within the course of a single research interview.

Reviewer: 2

1. In the INTRODUCTION, there are a number of statements that I would suggest should be supported by an academic reference. These include: '...increasing numbers of people are being diagnosed and this is set to rise further with our aging population'. 'Receiving a diagnosis of dementia.... physical and mental deterioration' 'Due to the nature of the illness....participate in conversations'.

References for these statements have been added (p.3). One new reference was added: Brayne C, Miller B. Dementia and aging populations—A global priority for contextualized research and health policy. *PLoS Medicine*. 2017;28:14(3 PubMed):e1002275.

2. It is also unclear what is meant by 'changes in competencies'

We have explained this in the text (p.3): (e.g. skills, autonomy)

3. In the second paragraph, the first sentence commencing with 'Traditionally, research has ignored...' is a very broad and sweeping statement and I do not agree....

We have removed this statement.

4. In the third paragraph, I think it is important to state that not only is dementia a chronic illness, but it is also a life-limiting illness.

As suggested, the statement has been changed to: "When receiving a chronic and life-limiting illness diagnosis such as dementia."

5. Also, it is unclear what is meant by the sentence commencing with 'The stigma is widespread....' – please make this clearer.

We have explained this in the text (p.3): This stigma is widespread, e.g. in physicians, patients and their families' perceptions and attitudes.[25]

6. You refer to 'small sample sizes at the bottom of page 3, as if it is a limitation. I would suggest that it is not, but rather the sample sizes are more indicative of the human nature of this research...

This statement has now been removed.

7. ..the obvious issue here is that the point of diagnosis does not always closely correlate to the point at which the dementia and its symptoms commenced. Some people may have symptoms often disregarded as 'ageing' for some time prior to receiving the formal diagnosis. I think this would be an important point to note.

In the results section the above point is demonstrated, e.g. "attributing forgetfulness to normal ageing helped to normalise dementia" (p.14). We have added the reviewer's suggestion in our discussion, (p.16): "This also demonstrates that for some people, symptoms are disregarded as normal ageing prior to receiving the formal diagnosis, and therefore the point of diagnosis does not always closely correlate to the point at which dementia symptoms commence."

8. At a clinic appointment for 'scheduled diagnostic feedback', does that mean individuals are given the news that they have dementia? if so, I would question whether it was appropriate to attempt recruitment for this study at the same time. How was 'capacity to consent' assessed by clinicians in the memory clinic?

The following text has been added (in addition to the information in the 'patient and public involvement' section): When the patient and their companion arrived at the clinic, a researcher approached them to discuss the study further and obtain written, informed consent. For patients without capacity to provide informed consent to participate (one patient in our sample) we followed the "Guidance on nominating a consultee for research involving adults who lack capacity to consent" (Department of Health 2008). The participants had mild to moderate dementia and their impaired cognitive capacities could lead to misunderstanding the study. Their companions were always present when the study was explained to them, and participants were able to opt out at any stage of the research, including the interview. As the interviews involved a discussion of the patient's illness, which may be distressing, it was made clear to them that they did not have to discuss topics that they didn't wish to.

9. Was a list of questions or guiding prompts used to guide the interviews? Were they structured, semi-structured or unstructured? There is also growing evidence for the use of a conversational approach to interviewing people with dementia as it is less confronting and preserves dignity. AND - It is difficult to comment on the findings when it is not clear what the questions were or how interviews were conducted.

Text has been added in the methods: 'Semi-structured, in-depth Interviews'. More information is also added in the new 'patient and public involvement' section.

10. In the RESULTS section, please also report SD for average age. Can you please explain why you had 61 participants but only 6 completed the cognitive test? Is the cognitive test part of this study specifically, or part of the larger study? If it is not part of this study, then do not report that here – suggest you refer to it in the Inclusion criteria. AND - In Table 2 you refer to MMSE, ACE-III and MiniACE. Each of these needs to be explained in the body of the paper. But as suggested previously, if these were used as a criteria for inclusion, then they should be detailed there, not in the results.

Text and references have been added to the results (p.5) to explain that this was part of routine assessment and administered by doctors: Doctors administered either the Mini Mental State Exam [] or Addenbrooke's Cognitive Examination III [] as part of routine assessment.

1 patient had not completed these tests (unknown reason). We report SD for age in the table.

11. In Table 1, not sure what is meant by 'further education'?

The following explanation was added: (education below degree level)

12. The conclusion is a bit brief. There is scope to make this more punchy, and clearly articulate the implications and 'so what' of this work... also articulates the future directions for research and how clinical practices associate with dementia care should change as a result, AND Reviewer 3: The conclusion of the paper that receiving a diagnosis is a confronting turning point in peoples lives seems to be too strong in the light of the findings that about one third of the participants were not surprised and/or less affected.

The conclusions in the abstract and the main text have been revised. Future research suggestion have been added to the conclusion (now 'conclusion and future research')

Receiving a diagnosis of dementia is often a turning point in peoples' lives. People are confronted with and frustrated by decreasing competence in everyday normal activities. They feel vulnerable and

in limbo not knowing what further deterioration they will experience and when this will happen. People are worried about disclosing their diagnosis to others, with some actively concealing their diagnosis even from close family. Limited time in diagnostic appointments and limited pre-diagnostic counselling and post-diagnostic support means people have few opportunities to address the emotional impact of being told one has dementia. Future research could address how to optimize participation of people with dementia in diagnostic feedback meetings, e.g. through spending more time orientating people to the purpose of the meeting and addressing the emotional impact of a diagnosis of dementia. The stigma experienced by patients could be explicitly addressed to mitigate the threat to one's self- and social identity. Finally, professionals could discuss the benefits of staying positive and helpful coping strategies used by others along with accepting support to live well with dementia.

Future research could address how to tailor the diagnostic process to people with fluctuating cognition, e.g., re-orientating people to the purpose of meetings during the diagnostic process, addressing the emotional impact and stigma of a life-changing diagnosis such as dementia and providing hope by focusing on how to live well with dementia. Further research can also explore the relationship between participants' socio-demographic factors (e.g. age, education) and attitudes/concerns regarding cognitive decline and receiving the diagnosis.

Reviewer: 3

1. The authors touch upon the loss of the abilities to drive. This is a complicated topic, and bringing this up in the disclosure visit may have practical consequences, which directly impact patients' experiences. I wondered if the authors could elaborate a bit more on this.

Additional text to highlight this has been added in the discussion (p.14), where we discuss loss of competencies and the diagnostic agenda): Similarly, people were worried about the impact of cognitive impairment on driving (e.g. losing the ability to drive/driving licence).

2. The first theme ("dissonance") seems to be a bit heterogeneous, as it encompasses quite some contrasting themes (illness vs normal aging, terrified vs. less affected). This seems to be a bit ambiguous, which seems to be somewhat unusual. Perhaps the authors can explain.

Thank you for this suggestion to refine the first theme. This has now been changed to: 'Accommodating to the diagnosis: threat to self and social identity through increasing visibility of dementia'

3: The paper may benefit from considering the diagnostic disclosure more as an interactive process between the clinicians and does for instance not describe the way the diagnosis was actually being disclosed, e.g., as bad news, an illness, as a handicap for which support...

The following text has been added (p.17):

Receiving a diagnosis of dementia is a communicative process. Peoples' subjective experience of cognitive decline prior to receiving a diagnosis is relevant to this process. Adding to this, are expectations of how others might react to the diagnostic label e.g. becoming stigmatised/excluded. In addition, we saw how dementia differs from other progressive diseases as people experience 'cognitive fluctuations' (interruption in the ongoing flow of awareness) []. These factors influence how the diagnosis is received and accommodated, and has implications for the diagnostic process, including pre-diagnostic counselling, and post-diagnostic support. Support prior to receiving the diagnosis would not only make the process less confusing for the patient (Samsi et al., 2014), but also help people who care for them, as due to the nature of the illness, it is typically carers who initiate assessments for possible dementia [Bunn, 14]. It would also prepare people for this diagnosis, as we found that over a third of the participants did not expect the diagnosis and many did not know what

dementia is. However, only one of the nine participating memory clinics, had a formal process in place for pre-diagnostic counselling, despite recommendations that this should be part of the diagnostic process (Hodge et al., 2014). In addition, despite the benefits of post-diagnostic support for adjusting to the diagnosis and enabling patients and their families to plan for the future and increase independence [Kelly 2016 32], there is still lack of adequate post-diagnostic support (Brayne, 2017, Vince et al., 2017).

VERSION 2 – REVIEW

REVIEWER	Mark Schweda Department of Health Services Research, School of Medicine and Health Sciences, University of Oldenburg
REVIEW RETURNED	08-Feb-2019

GENERAL COMMENTS	I think the revisions have helped to further improve the paper.
---

REVIEWER	Melissa Bloomer Deakin University
REVIEW RETURNED	31-Jan-2019

GENERAL COMMENTS	Thank you for the opportunity to review this manuscript again. I appreciate the authors' attempts to address the comments of three reviewers. I have some further recommendations for changes to improve the manuscript. I note that not all of my recommendations to the first version of your manuscript were addressed or at the very least acknowledged. Given the international audience of the journal, I would recommend that the manuscript be modified to suit. This means starting with the global and then the local:-  • UK should be written as United Kingdom the first time • Readers will need more context to understand that is meant by the 'Prime Minister's Challenge on Dementia 2020' • In the Methods section, the authors refer to London and Devon, but then in Table 1 it states London and Exeter. Suggest you replace the place names with 'metropolitan' and 'rural' or similar throughout and consistently. I would also ask that the authors provide a definition for Dementia and for Alzheimer's Disease early on in the manuscript. I would also suggest that the authors detail how a diagnosis is made (which test is considered diagnostic internationally) In the background section, thank you for attempting to provide an explanation for 'memory clinic' but the statement commencing 'In some areas of the UK..... take place on the same day' is irrelevant to the international audience who do not know where Devon is located. Suggest a rewording to simply state that some memory clinics are able to facilitate a same-day assessment and diagnostic service, others not. The authors also state that 'there have been recommendations for providing pre-diagnostic counselling and post-diagnostic support', but recommendations do not necessarily mean implementation – please elaborate. I would have also thought that pre-diagnostic counselling may be the remit of the referring clinician, such as a GP. Furthermore, if a person continues to attend the memory clinic, surely post-diagnostic support is inherent in care? Middle of page 4, please replace 'there's' with 'there has'. I also not a mix of American and UK English throughout that should be
--

	addressed. Bottom of page 4, what is the 'guidance' referred to in the statement commencing with 'In line with this, guidance has emphasised...'? Top of page 5, I think it is important to note that one reason why there may be limited research in this area is because many people are past the point of 'early' or 'mild' dementia at the point of diagnosis. Even your data shows that participants scored as low as 16 on the MMSE, suggesting some may have had dementia for some time prior to diagnosis. The data was collected as far back as 2014 so I am concerned/curious as to why it has not been published before now Please explain how 'Clinicians assessed whether patients had the capacity to consent to participate' – using what measures, how, who decided? How was the decision consistently applied? What was the timeframe from revealing the diagnosis to invitation? I am concerned that people may not be emotionally prepared to be approached to participate in this research if they had literally just received the diagnosis. If the invitation to participate came BEFORE the diagnosis, then it was a bit of a 'give away' that bad news was to come. Also, what was the rationale behind the two week timeframe from initial agreement to participate and follow up phone call? Is there evidence to suggest this timeframe was most appropriate? What other supports might the person (and their family) have in place to aid their coping during this time? My concern is for their welfare first and foremost. Can you perhaps detail this in terms of 'post-diagnostic support'? Please explain the following sentence, which seems incomplete 'The participants had mild to moderate....could lead to misunderstanding the study'. Please add a reference to 'Alzheimer's Society Research Network' and explain how this network informed the development of the interview guide. Was it based on evidence, or anecdotal feedback? Who in the network? Given interviews were video recorded, what was done with the video component? Thank you for including the section titled 'patient and Public involvement'. My main concern is that the involvement seems tokenistic, and does not really exemplify the inclusion of consumers throughout the research process Table 1, as suggested earlier, amend to 'metropolitan' and 'rural'. Also, for the purposes of this study, is it necessary to differentiate between the various diagnoses? - If they are all listed, then I feel it is incumbent upon the authors to define each. Each of the cognitive tests needs to be explained in terms their diagnostic use and indication of severity. What do the codes such as (P07) and (P102) mean? At first I thought it was participant number, but there are not that many participants. Is it page number – if so page numbers for what? In the Discussion, what is meant by 'The experiences reported in this paper should be understood as products of the contingencies of the interview situation'? Do you mean the results are situational, reflective of coping and emotions at the time, and may change over time? 'In line with previous research' is used a few times and is superfluous.
--	---

REVIEWER	Frans Verhey Maastricht University/ Alzheimer Centrum Limburg
REVIEW RETURNED	05-Mar-2019
GENERAL COMMENTS	The authors have adequately addressed the issues raised in my review. I have no further comments

VERSION 2 – AUTHOR RESPONSE

Reviewer 2 comments

UK should be written as United Kingdom the first time.

This has been corrected in the title, abstract, and in the introduction line 1: In the United Kingdom (UK)

Readers will need more context to understand that is meant by the 'Prime Minister's Challenge on Dementia 2020'.

Text has been changed to explain the focus which is raising awareness: efforts to raise awareness such as the Prime Minister's national challenge to fight dementia [2],

In the Methods section, the authors refer to London and Devon, but then in Table 1 it states London and Exeter. Suggest you replace the place names with 'metropolitan' and 'rural' or similar throughout and consistently.

We have now replaced this with urban and rural throughout the paper.

I would also ask that the authors provide a definition for Dementia and for Alzheimer's Disease early on in the manuscript.

The following definition has been added in the introduction: Dementia is a progressive condition, that describes a set of diseases that damage the brain and include 'memory problems, changes in mood and behaviour, and communication and reasoning problems' [2]. Alzheimer's disease is the most common cause of dementia [2].

I would also suggest that the authors detail how a diagnosis is made (which test is considered diagnostic internationally).

On page 4, we briefly describe the memory clinic process. The following text has been to provide more information on the cognitive tests used: using standardised instruments, e.g. the Mini Mental State Examination (MMSE) [6] or the Addenbrooke's Cognitive Examination III (ACE-III).[7] Information about these tests is also added in Supplementary table 2: Cognitive tests.

In the background section, thank you for attempting to provide an explanation for 'memory clinic' but the statement commencing 'In some areas of the UK..... take place on the same day' is irrelevant to the international audience who do not know where Devon is located. Suggest a rewording to simply state that some memory clinics are able to facilitate a same-day assessment and diagnostic service, others not.

Thank you for your suggestion. We have reworded as suggested: Some memory clinics facilitate a same-day assessment and diagnostic service, however in most clinics this takes place over one or two months.

The authors also state that 'there have been recommendations for providing pre-diagnostic counselling and post-diagnostic support', but recommendations do not necessarily mean

implementation – please elaborate. I would have also thought that pre-diagnostic counselling may be the remit of the referring clinician, such as a GP. Furthermore, if a person continues to attend the memory clinic, surely post-diagnostic support is inherent in care?

The following has been added in the introduction: However patients, carers and professionals express concerns about support before and after the diagnosis.[9, 10]

This is also mentioned in the discussion: “there is still lack of adequate post-diagnostic support [74-75].”

Middle of page 4, please replace ‘there’s’ with ‘there has’. I also not a mix of American and UK English throughout that should be addressed.

Many thanks for pointing this out. We have now corrected this and other instances, e.g. p.17 replaced ‘optimize’ with ‘optimise’.

Bottom of page 4, what is the ‘guidance’ referred to in the statement commencing with ‘In line with this, guidance has emphasised...’?

This has now been replaced with the World Health Organization, to specify the guidance.

Top of page 5, I think it is important to note that one reason why there may be limited research in this area is because many people are past the point of ‘early’ or ‘mild’ dementia at the point of diagnosis. Even your data shows that participants scored as low as 16 on the MMSE, suggesting some may have had dementia for some time prior to diagnosis.

Many thanks for your comment, which has now been added p.5: An important implication for research at the time of diagnosis is also the fact that many people may have had dementia for some time prior to diagnosis.

The data was collected as far back as 2014 so I am concerned/curious as to why it has not been published before now.

This was a large project which was finalised/closed in 2017. There was a large amount of data to be analysed and we also wanted to allow time for feedback from patient/carers and clinicians on the findings before finalising the analysis and writing a paper (first submitted Aug 2018). Initial findings from the patient interviews were presented at a conference (26th Alzheimer Europe Conference, Denmark 2016) and then were finalised in meetings with the research team and with lay members: people with dementia and a carers, at a workshop (2017). We also sent a summary of the study findings (April 2018) to those participants who had indicated in the consent form that they wanted to receive one. We waited for any feedback before we finalised and submitted this work in 2018.

Please explain how ‘Clinicians assessed whether patients had the capacity to consent to participate’ – using what measures, how, who decided? How was the decision consistently applied?

Assessing patient capacity was assessed as per local policy at each participating Mental Health Trust, who follow the ‘Mental Capacity Act (MCA) 2005’.

What was the timeframe from revealing the diagnosis to invitation? I am concerned that people may not be emotionally prepared to be approached to participate in this research if they had literally just received the diagnosis. If the invitation to participate came BEFORE the diagnosis, then it was a bit of a ‘give away’ that bad news was to come.

On p.5. we explain that Information sheets were sent with patient appointment letters, prior to their diagnostic meeting with the clinician. The information sheets did not reveal ‘bad news’ or any additional information regarding their meeting/potential diagnosis, to that of the clinic appointment letters.

Also, what was the rationale behind the two week timeframe from initial agreement to participate and follow up phone call? Is there evidence to suggest this timeframe was most appropriate? What other supports might the person (and their family) have in place to aid their coping during this time? My concern is for their welfare first and foremost. Can you perhaps detail this in terms of 'post-diagnostic support'?

We wanted to capture peoples' experiences soon after the diagnostic feedback meeting but allowing enough time for people to adjust after the meeting and make arrangements to visit people in their homes.

Please explain the following sentence, which seems incomplete 'The participants had mild to moderate....could lead to misunderstanding the study'.

The following text has been added to explain this: their involvement in the study.

Please add a reference to 'Alzheimer's Society Research Network' and explain how this network informed the development of the interview guide. Was it based on evidence, or anecdotal feedback? Who in the network?

We explain that the Alzheimer's Society is a 'dementia support and research charity' (p.7). and new text has been to explain who helped to design the forms: Research Network volunteers (group discussions), who have personal experience of dementia - living with the condition or as a carer or former carer.

Given interviews were video recorded, what was done with the video component?

The video component was not analysed separately. This was converted to audio and video files deleted. Audio files are stored on a secure/protected servers, as per Ethics agreement.

Thank you for including the section titled 'patient and Public involvement'. My main concern is that the involvement seems tokenistic, and does not really exemplify the inclusion of consumers throughout the research process.

We agree that additional involvement would have been ideal, however this was not possible due to time restrictions and having similar involvement of other stakeholders (clinicians). People living with dementia, carers/former carers were involved in the design of study/study documents (e.g. lay summaries and participant information) and also commented on the findings, which helped to contextualise results and shaped our interpretation if these – new text: Feedback helped to contextualise/interpret findings.

Table 1, as suggested earlier, amend to 'metropolitan' and 'rural'. Also, for the purposes of this study, is it necessary to differentiate between the various diagnoses? - If they are all listed, then I feel it is incumbent upon the authors to define each.

Table has been amended to 'urban' and 'rural' . Each dementia diagnosis is now defined in Supplementary files, table 1: Participants' diagnosis/dementia type description

Each of the cognitive tests needs to be explained in terms their diagnostic use and indication of severity.

Information to explain these has been added in Supplementary table 2: Cognitive tests.

What do the codes such as (P07) and (P102) mean? At first I thought it was participant number, but there are not that many participants. Is it page number – if so page numbers for what?

These are indeed participants numbers. People who participated in the larger - ShareD study, were asked if they would also like to be interviewed. 215 people participated in the ShareD study, of those only the ones who had received a diagnosis of the dementia were asked to be interviewed. 61 (not consecutive) agreed/were able to be interviewed. We did not change the original participant

numbers so that we could trace back to the original transcript and also to be able to link with participant numbers in other databases. We have added the original number of participants from the ShareD study in the methods: (n=215; consent rate 51%). In addition, a footnote on page 8, where participant number is first mentioned has been added explain this: Indicates participant number.

In the Discussion, what is meant by 'The experiences reported in this paper should be understood as products of the contingencies of the interview situation'? Do you mean the results are situational, reflective of coping and emotions at the time, and may change over time?

Thank you for pointing out that this was not clear. The following text based on your suggestion has been added (p.15): that is, findings are situational and reflect accounts and emotions at the time of the interview which may change over time.

'In line with previous research' is used a few times and is superfluous.

Thank you for your suggestion, this has been removed in some sentences and is now only mentioned twice in the discussion.